# Classification of the Burrower Bugs (Hemiptera: Heteroptera: Cydnidae): A Never-Ending Story?

**DOI:** 10.3390/insects16090874

**Published:** 2025-08-22

**Authors:** Jerzy A. Lis

**Affiliations:** Institute of Biology, University of Opole, Oleska 22, 45-052 Opole, Poland; cydnus@uni.opole.pl

**Keywords:** Pentatomoidea, Cydnidae, ‘cydnoid’ complex, classification history, classification framework

## Abstract

The family Cydnidae, commonly known as burrower bugs, is among the least studied in terms of phylogeny within Pentatomoidea. It remains an inadequately defined group, with its subfamilial and tribal structure varying over time depending on the author’s perspective. The current classification of the family is highly controversial, leading to numerous misinterpretations. In the broadest family concept, Cydnidae includes all taxa that are morphologically characterized by an array of more or less flattened setae forming coxal combs. This inclusive approach has resulted in the use of the term “Cydnidae *sensu* lato”, also known as the “cydnoid complex,” which encompasses taxa such as Cydnidae *sensu stricto*, Thyreocoridae, Parastrachiidae, and Thaumastellidae. At the opposite end is the family called “Cydnidae *sensu stricto*,” or simply “Cydnidae.” Under this scheme, the family is divided into six subfamilies: Amaurocorinae, Amnestinae, Cephalocteinae, Cydninae, Garsauriinae, and Sehirinae, while Parastrachiidae, Thaumastellidae, and Thyreocoridae are regarded as separate families. However, studies using morphological characters and molecular analyses have not yet reached a consensus on the taxonomic classification of Cydnidae. Further, more detailed research is required to resolve this issue.

## 1. Introduction

The family Cydnidae, the phylogeny of which remains poorly understood and the subject of considerable controversy, is often regarded as a key taxon for understanding the relationships and origins of the superfamily Pentatomoidea. The Cydnidae remains an inadequately defined group, and its subfamilial and tribal structure has varied over time depending on the author’s viewpoint. Contemporary classification within Pentatomoidea has proven to be highly controversial, leading to ongoing issues and several misinterpretations. As a result, it is currently impossible to specify the exact number of genera and species in the family, since several taxa that some authors regard as subfamilies of the Cydnidae (such as Parastrachiidae, Thaumastellidae, and Thyreocoridae) are often considered separate families within the Pentatomoidea.

Consequently, the number of genera and species assigned to the Cydnidae has varied (depending on the classification) from approximately 90 to over 144 genera and from approximately 700 to at least 1190 species [1,2,3,4,5,6,7,8,9,10,11,12].

In the broadest family concept, the Cydnidae includes all taxa that are morphologically characterized by an array of more or less flattened setae forming the coxal combs, a feature not found elsewhere in the Heteroptera, e.g., [12,13,14]. This inclusive approach has led to the use of the term “Cydnidae *sensu* lato”, also known as the “cydnoid complex” (a term first used by Lis [2], which encompasses the following taxa: Cydnidae *sensu stricto*, Thyreocoridae, Parastrachiidae, and Thaumastellidae, e.g., [2,12,13,14,15,16]).

At the opposite end of the approaches is the family called “Cydnidae *sensu stricto*” or simply “Cydnidae”. Under this scheme, the family is divided into six subfamilies: Amaurocorinae, Amnestinae, Cephalocteinae, Cydninae, Garsauriinae, and Sehirinae, while the Parastrachiidae, Thaumastellidae, and Thyreocoridae are regarded as separate families, entirely distinct from the Cydnidae, e.g., [9,10,11,14,17,18,19,20,21,22,23].

## 2. History of the Classification of Cydnidae

The history of the Cydnidae begins with Fabricius’ description of the genus *Cydnus*, the type genus of the family [24]. The family initially included 16 species; today, only one of those originally described by Fabricius [24] remains in the genus *Cydnus* (namely, *Cydnus tristis* (Fabricius, 1775), which is now regarded as a synonym of *Cydnus aterrimus* (Forster, 1771)). All other species have since been moved to different genera within the Cydnidae or other families within the superfamily Pentatomoidea.

The history of the family classification began with Billberg in 1820 [25], who proposed the name “Cydnides” for the group of species included in the genus *Cydnus* by Fabricius in 1803 [24]. Amyot & Serville in 1843 [26] established a race “Spinipèdes” (which at that time corresponded to the group of species placed in the genus *Cydnus* by Fabricius [24] and described several new genera to encompass some of those species).

The same authors [26] divided the “Spinipèdes” into two groups based on the shape of the anterior tibiae: the “Cydnides” (with expanded anterior tibiae bearing strong spines) and the “Séhirides” (with slender anterior tibiae lacking strong spines). Stål in 1864 [27] Latinized both names and proposed that all of the genera be regarded as a separate family, “Cydnida”, and divided the family into two subfamilies: “Cydnida” and “Sehirida”. In the same year, Mayr [28] used the properly Latinized family name Cydnidae for the first time.

In the early 1880s, Signoret [29,30,31,32,33,34,35,36,37,38,39,40,41], in the initial taxonomic revision of the group, accepted its division into two subgroups; however, he ignored the names suggested by Stål [27] and adopted the terms proposed by Amyot & Serville [26], namely “Cydnides” and “Séhirides”. Signoret [29,30,31,32,33,34,35,36,37,38,39,40,41] classified the entire group as the subfamily “Cydnides” within the family “Pentatomides”.

For a long time, the only categories within the family were the two subgroups, “Cydnides” and “Séhirides,” until Hart in 1919 [42] established the tribe Amnestini for the genus *Amnestus* Dallas, 1851. However, in addition to the subfamily Cydninae, Hart’s Cydnidae included the subfamily Thyreocorinae, which was generally considered a separate subfamily within the family Pentatomidae at the time. Moreover, Hart divided the subfamily Cydninae into three tribes: Cydnini, Sehirini, and the newly defined Amnestini.

Froeschner [43], in his 1960 monograph on the Cydnidae of the Western Hemisphere, proposed the following intrafamilial classification: Amnestinae (raised from the tribal to the subfamilial level), Garsauriinae (a newly described subfamily), Scaptocorinae (a newly described subfamily), Cydninae, and Sehirinae.

Shortly afterwards, Wagner [44], in a study of the taxonomic significance of the genital structures in the family Cydnidae, downgraded the subfamily Scaptocorinae to the tribal level within the subfamily Cydninae and proposed the following classification of the family: (1) subfamily Cydninae with three tribes, Cydnini, Scaptocorini (new status), and Geotomini (a new tribe); (2) subfamily Sehirinae with two tribes, Sehirini and Amaurocorini (a new tribe); and (3) subfamily Corimelaeninae (also including the European genus *Thyreocoris* Schrank, 1801).

Nearly twenty years later, Dolling [13] proposed a much broader concept of the family Cydnidae using the coxal combs at the apices of the coxae and a strigil on the ventral side of the metathoracic wing as the two crucial characters defining this family. Dolling [13] recognized eight subfamilies: Amnestinae, Garsauriinae, Scaptocorinae, Sehirinae, Cydninae, Thaumastellinae (previously regarded as a separate lygaeoid or pentatomoid family; see Seidenstücker [45,46] and Štys [47]), Thyreocorinae, and Corimelaeninae (the latter having been considered earlier as an independent family in the Pentatomoidea or a synonym of Thyreocorinae; see McAtee & Malloch [48]). Although Dolling [13] believed that all the aforementioned taxa appeared to form a single evolutionary lineage unrelated to any other line of Pentatomoidea, he completely omitted the sehirine tribe Amaurocorini from consideration.

Several years later, Schaefer et al. [49] proposed that the Parastrachiinae (comprising a single genus, *Parastrachia* Distant, 1883) should be classified within the family Cydnidae, thereby defining the broadest concept of the family at the time. Under this framework, the family Cydnidae would comprise nine subfamilies: Amnestinae, Garsauriinae, Scaptocorinae, Sehirinae, Cydninae, Thaumastellinae, Thyreocorinae, Corimelaeninae, and Parastrachiinae. However, similar to Dolling [13], the tribe Amaurocorini was excluded from consideration [49]. Shortly afterwards, Jacobs [50] proposed Thaumastellidae as the sister group of all other taxa within Dolling’s Cydnidae [13], but Gapud [15] retained Thaumastellinae as a cydnid subfamily, together with Parastrachiinae and Corimelaeninae.

In his phylogenetic analysis of Pentatomoidea, Gapud [15] followed Dolling [13] and considered the presence of coxal combs as apomorphic, viewing their occurrence in the broadly defined Cydnidae as a unique innovation. For many years, this character was the most important criterion for differentiating the broadly defined Cydnidae; however, a recent study [23] demonstrated the independent origin of the coxal combs within the superfamily Pentatomoidea.

In contrast to Dolling [13], Gapud [15], and Schaefer et al. [49], Linnavuori [51], in his monograph on West, Central, and North-East African Cydnidae, treated Thyreocoridae and Thaumastellidae as families separate from Cydnidae.

Schuh & Slater, in their book on true bugs of the world [1], followed Jacobs [50] in classifying the family Cydnidae and assigned Thaumastellidae as a separate family. The monograph [1] included eight subfamilies within Cydnidae: Amnestinae, Corimelaeninae, Cydninae, Garsauriinae, Parastrachiinae, Scaptocorinae, Sehirinae, and Thyreocorinae.

At the end of the 20th century [3,52], the subfamily name Scaptocorinae Froeschner, 1960, was replaced by its older term, Cephalocteinae Mulsant et Rey, 1866, and the subfamily was divided into two tribes, Scaptocorini and Cephalocteini.

Only in the first decade of the 21st century was the tribe Amaurocorini of the subfamily Sehirinae, which had been completely overlooked for over 40 years (e.g., [1,14,15,49]), elevated to subfamilial rank based on the morphological structure of the spermatheca [17]. Moreover, two tribes, the Geotomini and the Sehirini, were recognized as non-monophyletic based on the same character [17].

Grazia et al. [53] proposed that the family Parastrachiidae should be included within a more broadly conceived Thyreocoridae; however, this suggestion was not supported by subsequent studies of egg structure and oviposition patterns [10].

The concept recognizing six subfamilies within the Cydnidae—namely Amnestinae, Amaurocorinae, Cephalocteinae, Cydninae, Garsauriinae, and Sehirinae—was approved and widely accepted at the time (e.g., [6,9,10,11,14,17,18,19]). In this context, three taxa—Parastrachiidae, Thaumastellidae, and Thyreocoridae—were considered as distinct families of the Pentatomoidea. However, Schuh & Weirauch [12] proposed an alternative classification scheme for the Cydnidae. They disregarded the results of the analyses from the publications mentioned above based on morphological characters, as well as the first comprehensive molecular analysis of species within the broadly defined Cydnidae [20]. By ignoring the results of nearly forty years of research (see Discussion), the authors adopted an overly broad approach to the family Cydnidae, including the families Parastrachiidae, Thaumastellidae, and Thyreocoridae as subfamilies.

Recently, several important papers based on molecular studies [21,22,23,54,55] have offered new insights into the classification of the ‘cydnoid’ complex and the family Cydnidae itself. These studies indicate that Parastrachiidae, Thaumastellidae, and Thyreocoridae should be considered separate from the family Cydnidae. However, the analyses of Roca-Cusachs et al. [56] and Luo et al. [57] suggest a clade consisting of Parastrachiidae + Sehirinae (the subfamily of Cydnidae).

The historical classification, as summarized by Rider et al. [19], discusses various groups currently included in the Cydnidae. Most important contemporary classification schemes of the ‘cydnoid’ complex of families are presented in Table 1. Characters used during various studies on the classification of the ‘cydnoid’ complex of families (morphological *versus* molecular characters) are summarized in Table 2.

## 3. Discussion

The concept of the family Cydnidae has evolved over the last six decades, becoming generally narrower or broader depending on studies by various authors. Cydnidae has included three [51], five [2,43,51], six [2,17,19,22,23], eight [1,13], or even nine [12,49] subfamilies (Table 1). However, recognizing six subfamilies within the Cydnidae, namely Amnestinae, Amaurocorinae, Cephalocteinae (including Cephalocteini and Scaptocorini), Cydninae (encompassing Cydnini and Geotomini *sensu lato*), Garsauriinae, and Sehirinae *sensu lato*, is currently accepted and widely supported (e.g., [2,6,9,10,11,14,17,18,19,21,22,23]).

Nevertheless, in the most important contemporary publication concerning Heteroptera (by Schuh & Weirauch [12]), the classification of the family Cydnidae was addressed in a way that completely ignored the latest research on this family. By overlooking nearly 40 years of research, the authors took an overly broad view of the family Cydnidae, including the families Parastrachiidae, Thaumastellidae, and Thyreocoridae as its subfamilies. They provided a morphological diagnosis for their broadly defined Cydnidae [12], including characters that were considered likely to suggest a common origin for the four families (Cydnidae, Parastrachiidae, Thyreocoridae, Thaumastellidae). Unfortunately, other studies have concluded that these characters either evolved independently or were plesiomorphic in nature.

The following subsections provide brief discussions of the characters considered diagnostic for Cydnidae, as defined by Schuh & Weirauch [12]. The headings of the subsections are placed in quotation marks to reflect the original wording used in their monograph.

### 3.1. “Antennae usually 5-segmented”

The four-segmented antennae found in most Heteroptera are regarded as plesiomorphic [53,58] and are part of the heteropteran body groundplan [59,60].

The five-segmented antennae found in species of the four families (Cydnidae, Parastrachiidae, Thyreocoridae, Thaumastellidae) cannot be considered a diagnostic character for this group. Firstly, five-segmented antennae occur not only in the Cydnidae *sensu* Schuch & Weirauch [12] but also in other heteropteran groups (especially within the superfamily Pentatomoidea), e.g., [7,19,59,60,61,62].

However, some genera belonging to the family Cydnidae, as defined by Pluot-Sigwalt & Lis [17], retain the plesiomorphic state of this character (i.e., four-segmented antennae with no secondary subdivision of the second segment, the pedicel). This includes, for example, two genera in the subfamily Cydninae, tribe Geotomini (*Adrisa* Amyot & Serville, 1843, and *Geopeltus* J.A. Lis, 1990), and four genera of the subfamily Cephalocteinae, tribe Scaptocorini (*Afroropus* J.A. Lis, 1999; *Atarsocoris* Becker, 1967; *Scaptocoris* Perty, 1830; *Schiodtella* Signoret, 1882) [2,3,43,63].

Therefore, for the Cydnidae *sensu* Schuh & Weirauch [12], the term “antennae usually 5-segmented” is not adequate to the facts and cannot be considered diagnostic.

### 3.2. “Head subquadrate, semicircular, somewhat explanate”

The head shape is relatively consistent within the Parastrachiidae and Thaumastellidae, although it is not always subquadrate [19,47,64]. For example, in the Thaumastellidae, the head tends to be more triangular in outline and is certainly not explanate [47].

Some species of the Thyreocoridae, especially those in the subfamily Corimelaeninae, also feature a more or less triangular head shape [48].

In contrast, within the Cydnidae *sensu* Pluot-Sigwalt & Lis [17], the head shape frequently differs significantly from that identified as a diagnostic feature by Schuh & Weirauch [12], exhibiting considerable variation [2,51,65]. Most members of this family have a head that is subquadrate, semicircular, and somewhat explanate. Still, in some taxa (or even entire subfamilies), the head can exhibit an entirely different shape. For example, in some species of Scaptocorini (Cephalocteinae) and Sehirini (Sehirinae), the head is almost triangular [2,51,65]. In these cases, it can even be extremely elongated, as in *Tacolus majusculus* (Schouteden, 1910) [51]. In members of the subfamily Garsauriinae, the head may be quadrate and more or less blunt at the front [2,51,66,67,68].

Furthermore, the head shape proposed by Schuh & Weirauch [12] as a diagnostic feature for the broadly defined Cydnidae (“subquadrate, semicircular, somewhat explanate”) also occurs in some species of other families within the Pentatomoidea, such as Aphylidae [19], Plataspidae [69,70], and Pentatomidae, for example, Menidini [71] or Sciocorini [69,72]. This suggests that the feature cannot be considered diagnostic for Cydnidae *sensu* Schuh & Weirauch [12].

### 3.3. “Ostiolar groove elongate”

The ostiolar groove or ostiolar canal (*peritreme* according to Kment & Vilímová [73]) is more or less elongated in most taxa of the Pentatomoidea (e.g., [2,43,44,49,53,61,73,74,75]). Furthermore, within the subfamilies of the broadly defined Cydnidae, the peritreme can also be short and rounded or occasionally, more or less reduced [2,43,51,73,75]. Therefore, the Cydnidae *sensu* Schuh & Weirauch [12] cannot be defined by an elongated peritreme.

### 3.4. “A plectrum of closely spaced sclerotized teeth present on ventral surface of metathoracic wing close to base of Pcu”

Schuh & Weirauch [12] defined the plectrum as a movable part of the stridulatory structure, in contrast to the stridulitrum, which is a stationary part of the stridulatory mechanism. However, in publications that explore the morphology of members of the family Cydnidae (and the entire Pentatomoidea), these terms have the opposite meaning (e.g., [11,47,53,76,77,78]). Furthermore, the metathoracic wing vein referred to as the postcubital vein (Pcu) by Schuh & Weirauch [12] was homologized with the first branching of the anal vein (A1) many years ago [79] and has been consistently recognized as such [11,53,77].

It is also important to note that the stridulitrum (the plectrum of Schuh & Weirauch [12]) is not only found in the families broadly classified as Cydnidae but also occurs in many other families of Pentatomoidea, including Canopidae, Cyrtocoridae, Megarididae, Plataspidae, Scutelleridae, Tessaratomidae, and Urostylididae [11,13,53,76].

Furthermore, Lis & Heyna [77] and Grazia et al. [53] have shown that the stridulitrum is absent in the Garsauriinae, as well as in species of the genus *Dismegistus* Amyot & Serville, 1843 (Parastrachiidae), making this character useless for defining the entire Cydnidae *sensu* Schuh & Weirauch [12].

### 3.5. “Comblike row of setae (coxal combs) present on distal margin of all coxae”

Previous studies [20,21,22,23,53] have indicated that coxal combs, once considered autapomorphies, are examples of convergent evolution. The arguments supporting the independent emergence of coxal combs in various groups within the Cydnidae *sensu* Schuh & Weirauch [12] have already been discussed in the literature cited and will not be repeated here.

### 3.6. “Tibiae spinose”

The legs of true bugs show significant morphological variation depending on the position (fore-, mid-, or hindleg) and the structures of specific parts (e.g., coxa, femur, tibia, and tarsus) [12]. However, most structural differences can be easily linked to a species’ lifestyle [80]. Notably, phylogenetically unrelated taxa can sometimes exhibit the same or very similar modifications of the leg structure [80].

Although only Cydnidae *sensu* Schuh & Weirauch [12] within the Pentatomoidea are known to have tibiae with more or less robust setae on their margins, similar tibial setae can occur in other heteropteran groups belonging to Pentatomomorpha, Cimicomorpha, and even Nepomorpha [13,80,81,82,83,84,85,86].

Among the Pentatomomorpha, the ground-dwelling Pentatomidae, specifically the Sciocorini and Strachini [13,62,74,80], or predatory Asopinae [85], possess tibial spines comparable to those observed in the broadly defined Cydnidae.

Within the Cimicomorpha, certain species of the Anthocoridae, especially those in the Lasiochilinae (tribe Lasiochilini), have males with foretibiae that feature a row of teeth or spines on their inner margin, similar to many species of the tribe Geotomini in the subfamily Cydninae. However, unlike the Geotomini species, where these spines are associated with a subterranean lifestyle, in Lasiochilini males (Anthocoridae), the spines have a sexual function [70]. Additionally, the more or less long, stout spines and setae on the foretibial margins of species in the Reduviidae (e.g., in Emesinae [83] and Harpactorinae [84]) are not adaptations for burrowing or digging in the ground as in most Cydnidae but rather, are linked to their raptorial function.

In the Nepomorpha, the foretibiae of adult Ochteridae, particularly in *Ochterus marginatus* (Latreille, 1804) (see Figure 1b in [86]), are equipped with the same type of stout setae as in some Cydnidae *sensu* Schuh & Weirauch [12], especially in the Thaumastellidae [45,72]. This type of stout setae, similar to cephalic chaetotaxy, evolved independently in certain Cydnidae and Ochteridae due to their shared life history and cryptic behavior [17,87].

Gapud [15] suggested that tibiae without spines in Pentatomoidea are the ancestral form, while those with distinct spines are derived characters. Grazia et al. [53] and Schuh & Weirauch [12] considered a row of stout setae on the lateral margins of the foretibia as a trait unique to Cydnidae, Parastrachiidae, Thaumastellidae, and Thyreocoridae. However, as discussed earlier, spinose tibiae can also appear in other heteropteran groups and may serve functions beyond digging or burrowing. Therefore, the foretibial setae and spines, which play various roles in Heteroptera, should be viewed as independently evolved features.

Furthermore, the number and morphology of spines and setae on the foretibial margins can vary considerably both within and between subfamilies of the broadly defined Cydnidae [2,5,7,9,11,13,14,29,30,31,32,33,34,35,36,37,38,39,40,41,43,44,48,51,63], and there is no consistent system concerning protibial setae and spines that can be used to define the Cydnidae *sensu* Schuh & Weirauch [12] in a phylogenetic sense.

### 3.7. “Tarsi 3-segmented, although sometimes absent”

Cobben [88] suggested that having all adult legs with two-segmented tarsi represented the ancestral state of Heteroptera. The evolution of adult legs has resulted in taxa with three-segmented tarsi, the main ontogenetic change in tarsal number in many unrelated heteropteran groups [88].

However, Weirauch et al. [58] proposed an alternative hypothesis for tarsal segmentation, suggesting that the most recent common ancestor (MRCA) of Heteroptera had a three-segmented tarsus. Therefore, two-segmented or undivided tarsi should be considered independent cases of multiple reduction.

In either case, three-segmented tarsi cannot be regarded as a diagnostic trait of the broadly defined Cydnidae. If the first hypothesis is correct, three-segmented tarsi in Cydnidae would have arisen from ontogenetic changes in unrelated Heteropteran groups. If the second hypothesis holds, having a three-segmented tarsus would be a plesiomorphic condition, as previously suggested [15,53]. Therefore, the presence of a three-segmented tarsi cannot serve as a diagnostic characteristic for the Cydnidae as defined by Schuh & Weirauch [12].

Additionally, the lack of tarsi in some species of burrowing bugs cannot be considered unique to this group of taxa. A secondary reduction in the tarsus also occurs in many other Heteroptera that have legs with non-locomotive functions but are not specialized for digging, such as raptorial or fossorial legs [58,80,88]. Therefore, such a reduction in the number of tarsi cannot be assumed to be characteristic only of the Cydnidae *sensu* Schuh & Weirauch [12].

### 3.8. “Abdominal trichobothria on sterna 3–7 oblique or longitudinal, on spiracular line”

The abdominal sterna of all members of the infraorder Pentatomomorpha, except for Aradidae, possess trichobothria; furthermore, the same trait is displayed in two families, Pachynomidae and Velocipedidae, within the infraorder Cimicomorpha [19,53,88,89,90,91,92,93,94,95].

The Pentatomoidea are distinguished from other members of the Pentatomomorpha by having a maximum of two trichobothria on each abdominal sternite [15,19,53,58,91,92,94,95,96,97,98,99,100]. However, the oblique or longitudinal arrangement of trichobothria on the third to seventh abdominal sterna not only helps identify the broadly defined Cydnidae as suggested by Schuh & Weirauch [12], but can also be observed in other taxa within the infraorder Pentatomomorpha, including many pentatomoid families [11,15,19,53,58,91,92,94,95,96,97,98,99].

Furthermore, the abdominal trichobothria on sterna 3–7 in the Cydnidae *sensu* Schuh & Weirauch [12] can be arranged not only in oblique or longitudinal pairs but also transversely, or may even be absent on sterna 3 and 4 [11,15,19,43,51,53,97,98,99].

The data above show that the number of trichobothria and their arrangement on sternites 3–7 within the Pentatomoidea vary greatly. Therefore, the broadly conceived Cydnidae cannot be defined based on the presence of “abdominal trichobothria on sterna 3–7 in an oblique or longitudinal arrangement, on spiracular line”, as stated by Schuh & Weirauch [12].

### 3.9. “Second abdominal spiracle placed on a differentiated, usually membranous anterolateral portion of sternum”

In nearly all Pentatomoidea, the part of the abdominal sternum that bears the second spiracle is covered by the metapleuron (e.g., [13,15,49,53]). This spiracle, in the species of the broadly defined Cydnidae, is located on the unpigmented and occasionally weakly sclerotised area at the anterior margin of the sternum [13,43,47,50,100]. This trait is not exclusive to the Cydnidae *sensu* Schuh & Weirauch [12]. The same feature was also observed in species of the subfamily Cyrtocorinae within the family Pentatomidae [53,101,102].

Moreover, the extent of depigmentation and desclerotisation of the sternum bearing the second abdominal spiracle is often difficult to interpret clearly. This is especially true when the spiracle is unpigmented and thus difficult to distinguish from the rest of the anterior part of the sternum, e.g., [13,53,101]. It therefore appears that further research is necessary to gain a comprehensive understanding of this character across all families within the superfamily Pentatomoidea; consequently, it cannot yet be regarded as diagnostic for the family Cydnidae, as proposed by Schuh & Weirauch [12].

### 3.10. “Nymphal scent glands present between abdominal terga 3–4, 4–5, and 5–6”

Nymphal dorso-abdominal scent glands between terga III–IV, IV–V, and V–VI are known in many families within the Heteroptera, including all Pentatomoidea and some Coreoidea [88,103,104,105,106,107,108]. Moreover, the structure of nymphal and adult dorso-abdominal scent glands in Cydnidae *sensu* Pluot-Sigwalt & Lis [17] is highly diverse [103,106,109]. Therefore, no single type of this structure can be said to characterize the group of families included by Schuh & Weirauch [12] in their broadly defined Cydnidae.

### 3.11. “Pygophore as in Figure 95.1F,G”; “Aedeagus as in Figure 95.1I–K,R,S”; “Parameres as in Figure 95.1L, T”

Male genital structures in the broadly defined Cydnidae are much more complex and variable than those exemplified by Schuh & Weirauch [12].

The data from various studies [2,15,43,44,47,49,51,53,64,101,110,111,112,113,114,115,116,117] suggest that there is no consistent pattern of genital structure within the subfamilies included in the family Cydnidae *sensu* Schuh & Weirauch [12]. Therefore, it is currently impossible to identify diagnostic features associated with male genital structures that could universally recognize the broadly defined Cydnidae.

### 3.12. “Female 8th ventral laterotergites fused”

In Heteroptera, the ventral laterotergites, also called paratergites, are separated from the dorsal laterotergites and form part of the connexivum [12,49,64,111,112,118,119]. The 8th ventral laterotergites are sometimes regarded as part of the female genital plates [49].

The findings of Schaefer et al. [49] suggest that the non-fused 8th ventral laterotergites in female genital plates should be incorporated into the basic plan of the Pentatomomorpha female genitalia. This plesiomorphic condition, where the eighth ventral laterotergites are not connected, is common in many representatives of the superfamily Pentatomoidea, including Acanthosomatidae [74], Aphylidae [53], Pentatomidae [53,74], Phloeidae [53], Scutelleridae [53,61], and Tessaratomidae [53].

It is important to note that even in some species of the Cydnidae *sensu* Pluot-Sigwalt & Lis [17] belonging to the Amnestinae, Cydninae (Geotomini and Cydnini), and Cephalocteinae (Scaptocorini), the 8th ventral laterotergites are not fused and remain distinctly separate [44,69,111]. Unfortunately, Schuh & Weirauch [12] did not take this fact into consideration. The authors described the broadly defined Cydnidae [12] as having the female 8th ventral laterotergites fused. However, it is essential to recognize that this character cannot serve as a diagnostic feature for the Cydnidae *sensu* Schuh & Weirauch [12]. The fused state of the female 8th ventral laterotergites occurs in many families besides the Cydnidae, regardless of their internal classification. This includes species of Lestoniidae [53], as well as some Pentatomidae, Plataspidae, and Scutelleridae [74,111].

In certain instances, however, it is possible to discern an intermediate condition in which the two laterotergites are in contact yet not entirely fused. This is evidenced by their connection, which manifests as a distinct furrow or rim, varying in degree of visibility, e.g., in Amaurocorinae [44], Corimelaeninae [44], Cydninae [111], Garsauriinae [53], Sehirinae [44], and Thyreocorinae [44]. Therefore, the fused female 8th ventral laterotergites cannot be a diagnostic character for the Cydnidae as defined by Schuh & Weirauch [12].

### 3.13. “Gonocoxa 8 large and broad”

The 8th gonocoxae (gonocoxites) are large, broad structures found in all subfamilies within the family Cydnidae *sensu* Schuh & Weirauch [12,13,49,53,112,114]. However, they also occur in this form in many taxa belonging to other families within the superfamily Pentatomoidea, including those in Acanthosomatidae, Lestonidae, Pentatomidae, Plataspidae, and Scutelleridae [44,61,70,74,111,118]. Therefore, the presence of this feature does not lead to the conclusion that it is exclusive to the family Cydnidae *sensu* Schuh & Weirauch [12].

### 3.14. “Spermathecae small with two flanges”

The structure of heteropteran spermathecae is diverse and often highly complex [110]. This morphological differentiation is particularly evident in species of the superfamily Pentatomoidea [17,61,70,71,72,120,121,122,123]. The typical pentatomoidean spermatheca is characterized by an intermediate structure usually well delimited by two flanges and featuring an unsclerotised, flexible zone [17]. This form of spermatheca is found only in certain taxa classified under the Cydnidae *sensu* Schuh & Weirauch [12], namely Thyreocoridae (Thyreocorinae and Corimelaeninae), Parastrachiidae, Cephalocteinae, Cydninae (Cydnini and Geotomini), and Sehirinae (Sehirini) [17]. However, in the subfamilies Amaurocorinae, Amnestinae, and Garsauriinae, one or both flanges are absent [17].

The simple tubular form of the spermatheca found in Amaurocorinae is unusual and aberrant within all Pentatomoidea [17]. This infrequent form also occurs in several families of Pentatomomorpha other than Pentatomoidea, in the Alydidae [124,125,126], Coreidae [127], Rhopalidae [128], and Lygaeidae [129,130,131].

Similar to Amaurocorinae, the spermatheca in Amnestinae and Garsauriinae displays several similarities with the spermathecae described for Lygaeoidea and Coreoidea, primarily in the absence of both flanges (Amnestinae) or only a proximal flange (Garsauriinae) and the structure of the intermediate part [17,110,127,132]. These data suggest that the presence of two flanges cannot be regarded as a synapomorphy in the Cydnidae.

## 4. Conclusions and Suggestions

To date, the internal classification of the family Cydnidae has primarily depended on morphological characters, and these have been variously interpreted from a phylogenetic standpoint.The presence of an array of more or less flattened setae forming the coxal combs, a feature not observed elsewhere in the Heteroptera, has been the most significant morphological character used to define the family (regardless of the approach taken). However, as has been demonstrated multiple times, the coxal combs cannot be regarded as a synapomorphy for the taxa within the Cydnidae, since the same structure appears in many taxa outside the superfamily Pentatomoidea.Molecular analyses have consistently shown that not all subfamilies within the family Cydnidae (in its broadest sense) are related or form monophyletic groups. This is particularly true for the family Thaumastellidae, which has been repeatedly demonstrated to be unrelated to other taxa included in the family Cydnidae (even in its broadest sense).Consequently, researchers currently studying this family’s classification will find it confusing, as almost every publication presents a slightly different version.To address this issue and clarify the classification of the family Cydnidae, research should be conducted to answer the following questions:
aWhat is the correct systematic position of the family Thyreocoridae in relation to all other subfamilies of Cydnidae, particularly regarding the subfamily Amaurocorinae?bShould the subfamilies Amnestinae and Garsauriinae be considered part of the family Cydnidae, or should they be classified as separate monophyletic groups outside this family?cShould Cephalocteinae be regarded as a distinct subfamily, or should it be included as a tribe within the subfamily Cydninae?dDo the genera currently included in the family Parastrachiidae (*Dismegistus* and *Parastrachia*) form a monophyletic group together with taxa in the subfamily Sehirinae?

Answering these questions would clarify the complex classification of the Cydnidae and its subfamilies and establish the systematic position of related families, such as the Parastrachiidae and Thyreocoridae. As research in this area is ongoing and the results are promising, answers to these questions are expected soon.

## Figures and Tables

**Table 1 insects-16-00874-t001:** Selected contemporary classification schemes of the ‘cydnoid’ complex of families.

Author(s), [Reference Number]	Families of the ‘Cydnoid’ Complex	Subfamilies	Author(s), [Reference Number]	Families of the ‘Cydnoid’ Complex	Subfamilies
Froeschner (1960) [43]	Cydnidae	Amnestinae Cydninae Garsauriinae Scaptocorinae Sehirinae	Dolling (1981) [13]	Cydnidae	Amnestinae Corimelaeninae Cydninae Garsauriinae Scaptocorinae Sehirinae Thaumastellinae Thyreocorinae
	Corimelaenidae	-			
Schaefer et al. (1988) [49]	Cydnidae	Amnestinae Corimelaeninae Cydninae Garsauriinae Parastrachiinae Scaptocorinae Sehirinae Thaumastellinae Thyreocorinae	Linnavuori (1993) [51]	Cydnidae	Cydninae Garsauriinae Sehirinae
				Thaumastellidae	-
				Thyreocoridae	-
Lis (1994) [2]	Cydnidae	Amnestinae Cydninae Garsauriinae Scaptocorinae Sehirinae	Schuh & Slater (1995) [1]	Cydnidae	Amnestinae Corimelaeninae Cydninae Garsauriinae Parastrachiinae Scaptocorinae Sehirinae Thyreocorinae
	Parastrachiidae	-		Thaumastellidae	-
	Thaumastellidae	-			
	Thyreocoridae	Corimelaeninae Thyreocorinae			
Pluot-Sigwalt & Lis (2008) [17]; Rider et al. (2018) [19]; Lis & Domagała (2024) [22]; Lis et al. (2024) [23]	Cydnidae	Amaurocorinae Amnestinae Cephalocteinae Cydninae Garsauriinae Sehirinae	Schuh & Weirauch (2020) [12]	Cydnidae	Amaurocorinae Amnestinae Cephalocteinae Cydninae Garsauriinae Parastrachiinae Sehirinae Thaumastellinae Thyreocorinae
	Parastrachiidae	-			
	Thaumastellidae	-			
	Thyreocoridae	-			

**Table 2 insects-16-00874-t002:** Characters used during the studies on the classification of the ‘cydnoid’ complex of families (COI—cytochrome oxidase I, 12S—mitochondrial gene encoding 12S rRNA, 16S—mitochondrial gene encoding 16S rRNA, 18S—nuclear gene encoding 18S rRNA, 28S—nuclear gene encoding 28S rRNA, 2D—secondary structure of LVR L of 18S rRNA, 3D—tertiary structure of LVR L of 18S rRNA).

Author(s), Year of Publication, and Reference Number	Characters Analyzed							
	Morphology	Molecular Markers						
		COI	12S	16S	18S	28S	2D	3D
Froeschner (1960) [43]	+	−	−	−	−	−	−	−
Wagner (1963) [44]	+	−	−	−	−	−	−	−
Dolling (1981) [13]	+	−	−	−	−	−	−	−
Schaefer et al. (1988) [49]	+	−	−	−	−	−	−	−
Linnavuori (1993) [51]	+	−	−	−	−	−	−	−
Lis (1994) [2]	+	−	−	−	−	−	−	−
Grazia et al. (2008) [53]	+	+	−	+	+	+	−	−
Pluot-Sigwalt & Lis (2008) [17]	+	−	−	−	−	−	−	−
Lis (2010) [14]	+	−	−	−	−	−	−	−
Lis et al. (2012) [54]	−	−	+	+	−	−	−	−
Lis et al. (2017) [20]	−	−	−	−	+	+	−	−
Lis (2023) [21]	−	−	−	−	+	−	+	+
Lis & Domagała (2024) [22]	−	−	−	−	+	−	+	+
Lis et al. (2024) [23]	+	−	−	+	−	−	−	−

## Data Availability

No new data were created.

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
