# Peer review of "Classification of the Burrower Bugs (Hemiptera: Heteroptera: Cydnidae): A Never-Ending Story?"

_insects, 2025, doi:10.3390/insects16090874_

Round 1
Reviewer 1 Report
Comments and Suggestions for Authors
This article is a major contribution to clarifying the classification of the family Cydnidae. The author, along with several co-authors, has been working with this family for the last decade, studying morphology and phylogenetic relationships, and has published approximately twenty articles on the subject.

Author Response
Comment 1: This article is a major contribution to clarifying the classification of the family Cydnidae. The author, along with several co-authors, has been working with this family for the last decade, studying morphology and phylogenetic relationships, and has published approximately twenty articles on the subject.
Response 1: Thank you very much for your positive opinion. I am incredibly grateful for the recognition of my extensive research on the family Cydnidae, encompassing both morphological and phylogenetic aspects. All the suggestions placed directly in the paper have been considered, and the text was amended accordingly.

Reviewer 2 Report
Comments and Suggestions for Authors
It's very interesting paper, the author discusses the family Cydnidae, which is one of the least studied phylogenetically within the Pentatomoidea and the current classification of the family is highly controversial, depending on the author's point of view.
This article provides a comprehensive review of the historical development of research on the classification of the family Cydnidae, and some directions for future research to confirm the monophyleticity of this family and its classification structure are proposed.
The chapter History of the Classification of Cydnidae is very well written, including a discussion of various authors' research on this family. The author referred to the results obtained in the various works. It is undoubtedly the most work-consuming and time-consuming part of the manuscript, and additionally indicates the author's excellent knowledge of the subject.
The most extensive section of the work is the discussion, in which the author discusses the features used to define the Cydnidae family, which have evolved in different ways. These include complexes of morphological and molecular traits.
The analysis of these traits led the author to conclude that the analyses carried out so far, both morphological and molecular, still have not resolved the systematic uncertainties within this family.
The analysis of these features led the author to conclude that the analyses carried out so far, both morphological and molecular, still have not resolved the systematic uncertainties within this family. They should still be carried out, and, what is an optimistic emphasis of this work, they are being carried out and will soon answer the questions that have been bothering heteropterologists. Although this work has not solved the classification problem within the family Cydnidae, it is very important scientifically because it clarifies the problem and directs research to the right paths, outlining the issues that should be resolved. I strongly recommend publishing this manuscript in Insects in its present form.
Author Response
Comments 1: It's very interesting paper, the author discusses the family Cydnidae, which is one of the least studied phylogenetically within the Pentatomoidea and the current classification of the family is highly controversial, depending on the author's point of view.
This article provides a comprehensive review of the historical development of research on the classification of the family Cydnidae, and some directions for future research to confirm the monophyleticity of this family and its classification structure are proposed.
The chapter History of the Classification of Cydnidae is very well written, including a discussion of various authors' research on this family. The author referred to the results obtained in the various works. It is undoubtedly the most work-consuming and time-consuming part of the manuscript, and additionally indicates the author's excellent knowledge of the subject.
The most extensive section of the work is the discussion, in which the author discusses the features used to define the Cydnidae family, which have evolved in different ways. These include complexes of morphological and molecular traits.
The analysis of these traits led the author to conclude that the analyses carried out so far, both morphological and molecular, still have not resolved the systematic uncertainties within this family.
The analysis of these features led the author to conclude that the analyses carried out so far, both morphological and molecular, still have not resolved the systematic uncertainties within this family. They should still be carried out, and, what is an optimistic emphasis of this work, they are being carried out and will soon answer the questions that have been bothering heteropterologists. Although this work has not solved the classification problem within the family Cydnidae, it is very important scientifically because it clarifies the problem and directs research to the right paths, outlining the issues that should be resolved. I strongly recommend publishing this manuscript in Insects in its present form.
Response 1: Thank you very much for your positive opinion. I am highly grateful for the recognition of my extensive research on the family Cydnidae, encompassing both morphological and molecular aspects.
Reviewer 3 Report
Comments and Suggestions for Authors
Good review. I have only very few suggestions for improvement, few more references should be added.
One general remark. In this paper the characters given by Schuh & Weirauch are rejected one by one. If these characters are understood as synapomorphies than the critisims is perfectly sound, but I think that their paper used them only as "usual" characters helping to distinguish Cydnidae from other families. It is true that three-segmented tarsi and five-segmented antennae are not synapomorphies, but are diagnostic when comapring the most members of Cydnidae with e.g. Acanthosomatidae.
Is there (or not) potentionally any other character, not listed by Schuh & Weirauch, which could be suggested as synapomorphy for cydnoid complex.

Author Response
Page 1, line 33:
Reviewer: I am sorry but to mee they seems the most studied group of Pentatomoidea. For Scutelleridae or Acanthosomatidae we have just one phylogenetic paper, for Plataspidae or Scutelleridae none. But it does not change the fact that the phylogeny of Cydnidae is still poorly known.
My response: Yes, I agree with your comment. The text was amended as follows (lines 33–35): “The family Cydnidae, the phylogeny of which remains poorly understood and the subject of considerable controversy, is often regarded as a key taxon for understanding the relationships and origins of the superfamily Pentatomoidea.”
Page 2, line 58:
Reviewer: too big gap.
My response: It was corrected.
Page 4, lines 154–157:
Reviewer: The final step of the story should be added: the analyses of Roca-Cusachs et al. (2022) and Luo et al. (2025) suggest a clade consisting of Parastrachiidae+Sehirinae.
My response: It was corrected according to the Reviewer's suggestion. Two papers were added and cited.
Roca-Cusachs, M.; Schwertner, C.F.; Kim, J.; Eger, J.; Grazia, J.; Jung, S. Opening Pandora's box: Molecular phylogeny of the stink bugs (Hemiptera: Heteroptera: Pentatomidae) reveals great incongruences in the current classification. Syst. Entomol. 2022, 41, 36–51. https://doi.org/10.1111/syen.12514.
Luo, J.-Y.; Wu, Y.-Z.; Kment, P.; Salomão, A.T.; Damken, C.; Wang, Y.-H.; Xie, Q. Origin of the only myrmecomorphic stink bug, Pentamyrmex spinosus (Hemiptera: Pentatomidae), in the radiation era of ants (Hymenoptera: Formicidae). Syst. Entomol. 2025, 50, 415–427. https://doi.org/10.1111/syen.12664
Page 5, Table 1:
Reviewer: As right and left part of the table are separate entities, they should be divided by firm line. Formating need some improvement.
My response: Table 1 in the original version of my MS sent to the Editors of INSECTS was divided by the vertical firm line. Moreover, changing the original table layout caused all the other formatting errors. So, I hope the Editors can fix it.
Page 5–6, Table 2: see above. Table 2 was also formatted initially with the correct layout.
Line 194: “secondary” was corrected to “tertiary”.
Page 7, subsection 3.4:
Reviewer: Here I would cite also the excellent review on strudulatory devices of Heteroptera by Davranoglou et al.
My response: The publication has been cited in the appropriate place in the text.
Davranoglou, L.R.; Taylor, G.K.; Mortimer, B. Sexual selection and predation drive the repeated evolution of stridulation in Heteroptera and other arthropods. Biol. Rev. 2023, 98, 942–981. https://doi.org/10.1111/brv.12938
Page 8, line 281
Reviewer: also Polididus (DOI: 10.5281/zenodo.181485)
My response: The publication has been cited in the appropriate place in the text.
Ishikawa, T.; Sumiartha, K.; Okajima, S. The second representative of the genus Polididus (Hemiptera: Heteroptera: Reduviidae) in Southeast Asia, with partial redescription of P. armatissimus. Zootaxa 2008, 1740, 45-52, DOI: 10.11646/zootaxa.1740.1.5
Page 9, line 302
Reviewer: in phylogenetic sense
My response: I have included this sentence in the text.
